

# Implicit detection of user handedness in touchscreen devices through interaction analysis

Carla Fernández[1], Martin Gonzalez-Rodriguez[1], Daniel Fernandez-Lanvin[1], Javier De Andrés[2] and Miguel Labrador[3]

[1] Department of Computer Science, University of Oviedo, Oviedo, Asturias, Spain
[2] Department of Accounting, University of Oviedo, Oviedo, Asturias, Spain
[3] Department of Computer Science, University of South Florida, Tampa, FL, United States of America

## ABSTRACT

Mobile devices now rival desktop computers as the most popular devices for web surfing and E-commerce. As screen sizes of mobile devices continue to get larger, operating smartphones with a single-hand becomes increasingly difficult. Automatic operating hand detection would enable E-commerce applications to adapt their interfaces to better suit their user's handedness interaction requirements. This paper addresses the problem of identifying the operative hand by avoiding the use of mobile sensors that may pose a problem in terms of battery consumption or distortion due to different calibrations, improving the accuracy of user categorization through an evaluation of different classification strategies. A supervised classifier based on machine learning was constructed to label the operating hand as left or right. The classifier uses features extracted from touch traces such as scrolls and button clicks on a data-set of 174 users. The approach proposed by this paper is not platform-specific and does not rely on access to gyroscopes or accelerometers, widening its applicability to any device with a touchscreen.

## INTRODUCTION

This paper proposes a classification device to determine the user operation hand (handedness) in order to help web developers to customize the user interfaces dynamically, thus improving the usability of their designs.

The proposed model, founded upon the use of machine learning algorithms, is based exclusively on data gathered by agents embedded inside E-commerce web applications, observing users while they spontaneously interact with their browser, as they normally would do in their own computational context. This allows the detection of the user's handedness after very few interaction actions with the system (such as scrolling or point and clicks). Once the users are classified as left-handed or right-handed, the user interface can be dynamically adapted to the specific interaction requirements of the users' handedness.

Corresponding author
Martin Gonzalez-Rodriguez,
martin@uniovi.es

This topic is important because as of 2019, there were about 5.112 billion unique mobile users in the world over a total population of 7.676 billion (*We Are Social, 2019*). Penetration rate of mobile technology in 2018 was of 68% and the number of smartphone mobile users increased in 100 million in the period January 2018 to January 2019 (*We Are Social, 2019*) .

However, this important penetration rate of mobile technologies is not usually supported by high levels of ease-of-use and/or accessibility. Mobile app usability related problems are the fifth most frequent user complaint (*Khalid et al., 2015*), while about 52% of mobile users experienced usability problems relevant enough to impact on their loyalty and/or trustworthy feelings about E-commerce sites (*Experienced Dynamics, 2015*). These usability problems not only downgrade the company brand mark but also increase the chances of customers looking for similar services in the competitors.

Usability and accessibility on mobile computing is usually affected by context-specific issues. Among other requirements, usability engineering must deal with a large variety of screen sizes and device shapes. The display layout created to hold the application's interactive objects (buttons, menu items, text, etc.) must take in consideration the hand posture required to operate the device, which is heavily influenced by the so-called user's handedness (*Azenkot & Zhai, 2012*).

Progressive increment in the display size of such devices introduced novel usability problems when the users try to use them with only one hand (*Guo et al., 2016*; *Löchtefeld et al., 2015*). These usability issues reveal to be even more relevant if we consider that 49% of the users like to use their mobile devices with their thumb (*Hoober, 2013*).

Operating a mobile device with only one hand may be difficult for some users. Figure 1 shows the difficulties experienced by users of large displays when they try to access certain areas with their thumb. The access to the top of the display and/or to the side opposite to the operating hand are some examples of difficult or annoying interactions. Usability problems arise when the application screen layout forces interaction on those areas since users are implicitly requested to modify their posture in an uncomfortable way. Even worse, this may also represent an accessibility problem for motor-disabled users. People with a limited range of movement in their hand or fingers may find hard (or even impossible) to reach some areas of the display, preventing them to use specific features of their mobile applications. This paper is focused on how to classify users implicitly and dynamically according to their operational hand.

Some applications provide enhanced user experience (UX) through customization. Customized interactive dialogues are created on demand to satisfy the interaction requirements of users running applications in a specific interaction context (*Gonzalez-Rodriguez et al., 2009*). The changes and adaptations on the layout of the user interface are implicitly available to the user only when the application is able to infer or detect the interaction context. However, most applications are not explicitly aware of changes in the context of interaction, so these must be explicitly reported by the user to manually activate the customization process (*Löchtefeld et al., 2015*). The explicit selection of the interaction context (e.g., changing from portrait to landscape display modes in a mobile device) may result in an annoying process to certain users.

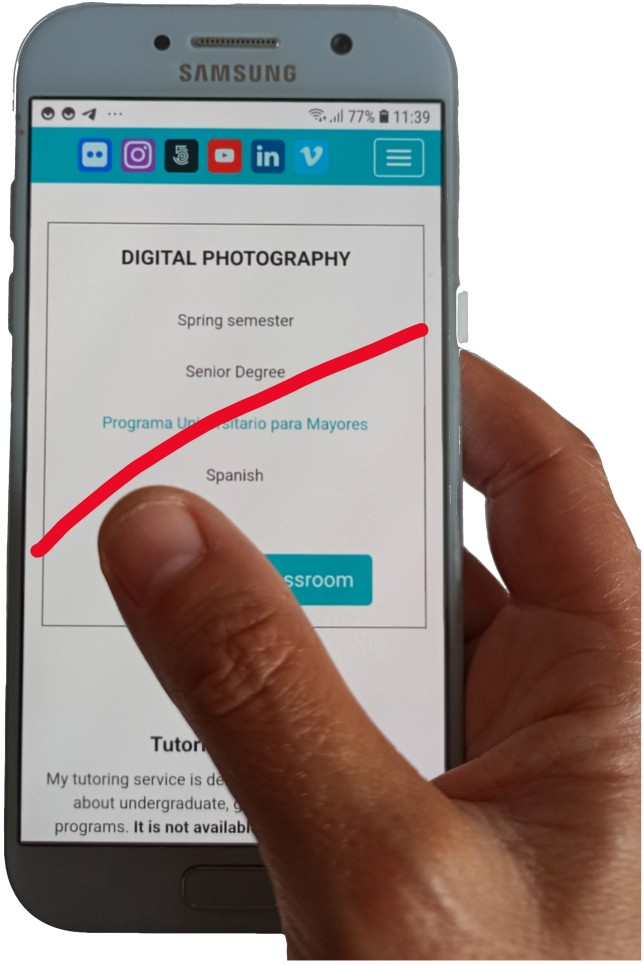

**Figure 1** **Red line shows the maximum thumb's motion range on a 5.4" size mobile display.** Source: own elaboration.

The customization of the user interface enhances the user experience when it is adapted to the specific interaction requirements of the user's operational hand (*Hussain et al., 2018*). The detection o user features can be done implicitly and/or dynamically. Implicitly means that users do not need to activate any option or preference in the interface in order to select their operational hand. They may not even know that such an option exists. Dynamically means that the system continuously monitors the interaction context looking for variations that force users to change their operational hand (eg. when users have to carry a heavy item with their operational hand, forcing them to use their mobile devices with the other hand). Such kind of system would help developers to detect the user's operational hand (left-handedness or right-handedness) at any moment, thus they would be able to apply the corresponding customized style to the display layout. To the extent we know, customization of mobile web interfaces to accommodate user handedness has been barely explored. Discussion about how this customization process can be done and its implications in terms of user interaction is beyond the scope of this research.

This paper proposes a model based on the use of machine learning techniques to classify users implicitly and dynamically according to their operational hand, reaching an accuracy of 99.92%. Unlike other approaches, where users are forced to execute specific predefined operations to facilitate the classification, the proposed solution encourages spontaneous interactions without limiting or guiding the kind of actions to be done. The ultimate goal is to avoid possible biases caused by non spontaneous behaviour (*Kaikkonen et al., 2005*).

Unlike other approaches, the proposal does not require reading data from the internal sensors of the mobile phone (accelerometers, gyroscopes, etc.), Therefore there is no additional battery consumption. It neither requires the installation of platform specific software (Android, iOS, Windows Mobile, etc.). Thus it can be used in any touchscreen based mobile device, provided that it is able to run a web browser. This approach also avoids the bias and reading noise specific to each device as the performance of gyroscopes and accelerometers varies significantly between different devices (*Kos, Tomažič & Umek, 2016*).

The paper is organized as follows. 'Thumb-Based Interaction Patterns' describes the so called "thumb zones" and their relevance for human computer interaction studies. 'Prior Research' presents the research background with a representation of the related studies on interface adaptation and algorithms designed to detect the user's operational hand. 'Design of the Study' describes the experimental design, the data gathering and depicts the sample distribution while 'Variables of the study' describes the analyzed variables. 'Statistical Methods' shows the machine learning strategies adopted to select the best-performing algorithm. 'Algorithm evaluation' discusses the performance of the most rated algorithms. 'Limitations' points out the research limitations while 'Conclusions and Future Work' discuss the results and describes the future research.

## THUMB-BASED INTERACTION PATTERNS

Human Computer Interaction defines the so-called "Thumb Zones" as areas of mobile displays that have the same easiness of access for the thumb (*Hoober & Berkman, 2011*). They are defined for both the left and right thumbs and are applicable independently of the user's laterality or handedness.

In accordance with the easiness of access, the display is divided into three areas (see Fig. 2). The easiest area to access for one-handed operation, and therefore the most comfortable area, is known as the "natural" area. It is the closest area to the user's thumb. The second area, the so-called "extent" area, entails some difficulties for the user to access it, but it is still usable. The last area, the so-called "hard" area, requires the user to modify the wrist position to enlarge the thumb operational swipe area. As a result, the access to elements located in that area is uncomfortable for most of the users and even painful or inaccessible for some others, specially for those users with motor disabilities and/or who interact with large displays (*Scott Hurff, 2014b*).

An example on how thumb zones affect usability is shown in Fig. 3. Overlays of these areas have been applied to the user interface of Facebook to show the degree of easiness for reaching relevant interactive objects. In the example, relevant menu options like

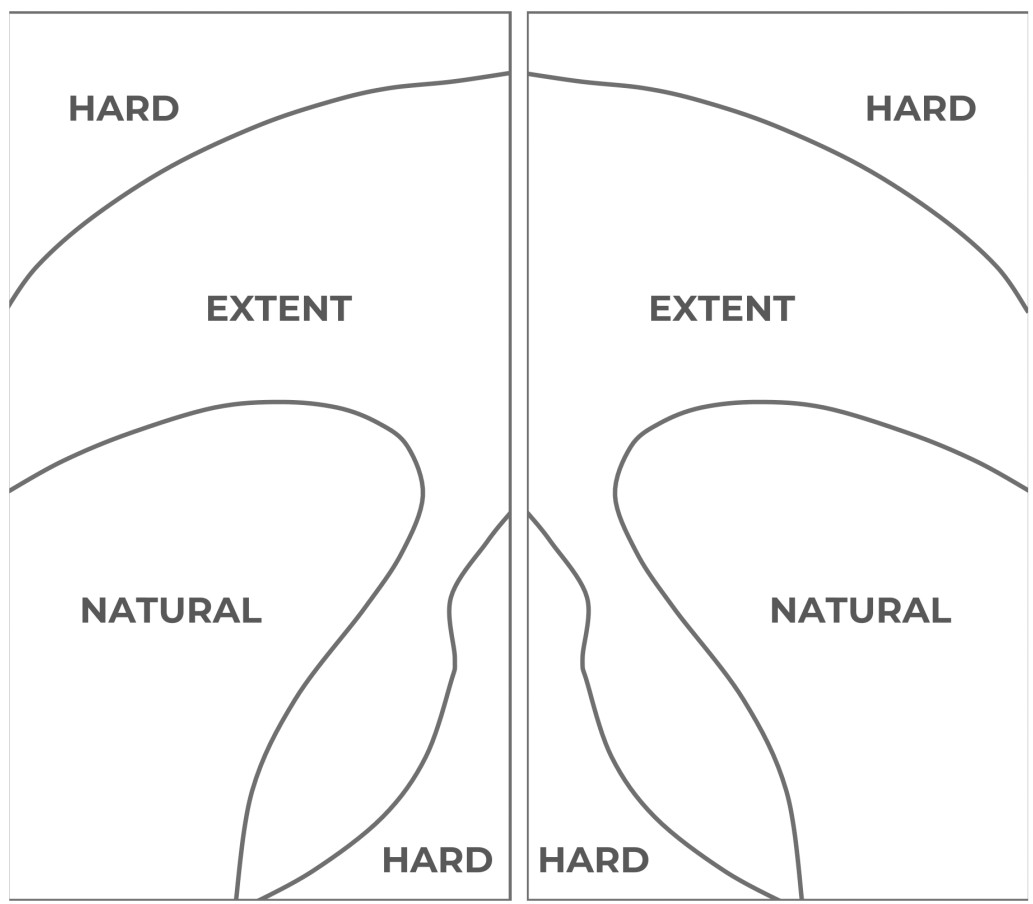

**Figure 2** **Thumb Zone mappings in a large 5.5" screen.** Left: thumb areas for a left-handed user. Right: thumb areas for a right-handed user. Source: adapted from *Scott Hurff (2014)*.

''Home'' or ''Events'' are hard to reach by left-handed users but they are easily accessible to right-handed users, even though these elements are not located in the right-handed users' comfort (natural) zone.

Since the thumb zones for left and right-handed users are different, the difficulty level required to access the same elements (buttons and menu items) will be different, thus their user experience and satisfaction will be different too.

Scrolling operations are another challenge to easiness of use related to handedness-based interaction on touchscreen displays. Most of the users start their scrolling actions placing the thumb in the area near to the center of the screen, swiping to create an arc-shaped trajectory that points to the side where the hand is holding the device (to the left, in the case of left-handed users or to the right for right-handed users) (*Goel, Wobbrock & Patel, 2012*). In the cited example, the location of the ''like'' button makes it easier to unwillingly click on it while scrolling through the display. Since clicking on this kind of button is done in a single step action, not requiring extra confirmation dialogues, the erroneous action may be executed implicitly, thus may be annoying to some users, thus diminishing the user
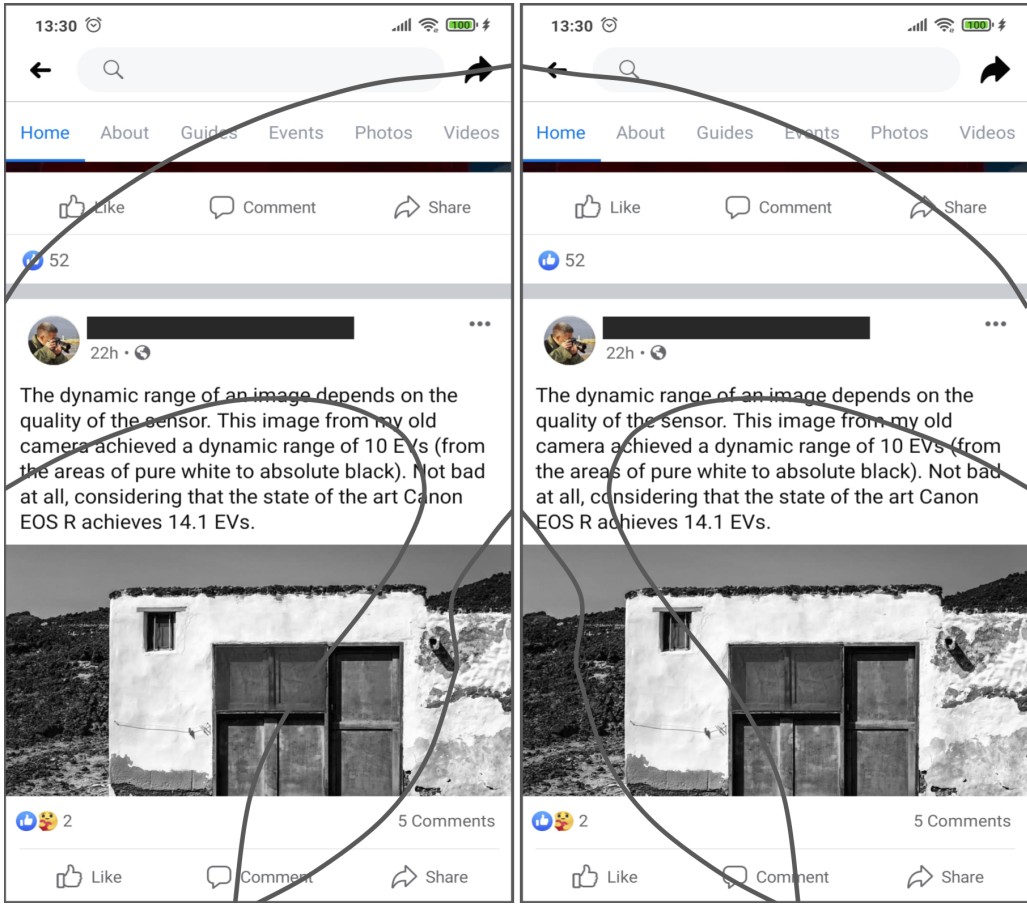

**Figure 3** **The thumb-zones described in Fig. 2 are overlaid over the user interface of the Facebook app for Android.** Relevant interaction objects like the 'share' or 'home' buttons are hard-to-reach for left-handed users. On the other hand, elements like the 'photos' or the 'like' buttons are hard-to-reach for right-handed users. Source: own elaboration.

experience. In an analogous way, right handed users may be prone to unintentionally click on the "share" button. Figure 4 shows a possible adaptation of the user interface, designed to comply with the interaction requirements of both left-handed and right-handed users.

## PRIOR RESEARCH

Adaptation of user interfaces to the user handedness on mobile devices is mostly focused on improving the performance in touchscreen operations with one hand only. The process explores different locations and size to locate interactive objects. Studies on user handedness in Human Computer Interaction have been mostly focused on human performance. *Khan & Rizvi (2009)* studied the influence of handedness on data entry performance in typing tasks, while *Shen et al. (2017)* measured the performance differences in keystroke tasks to attempt handedness recognition in computer forensics analysis.

*Parhi, Karlson & Bederson (2006)* determined the optimal target size area when using a mobile application with thumb only. Along the tests, 20 right-handed volunteers were

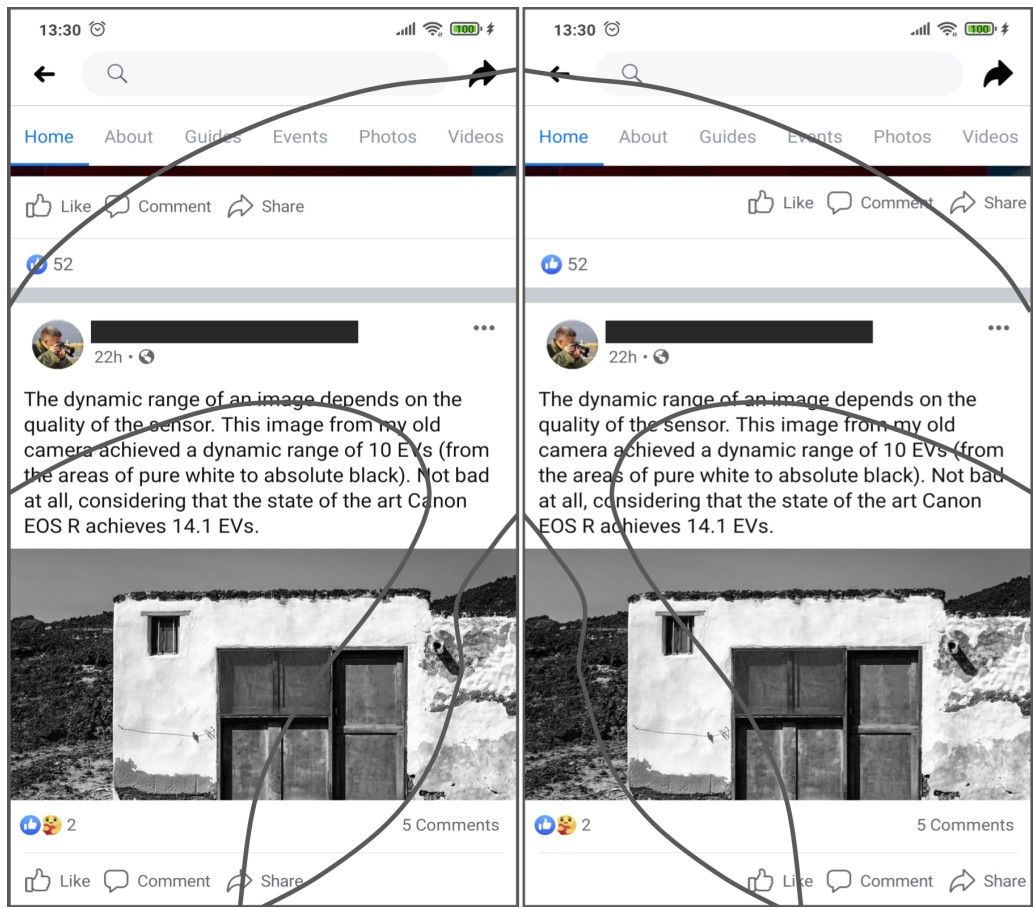

**Figure 4** **Example of an interface adaptation designed to make the interactive objects cited as an example in Fig. 3 more usable.** The 'share' and 'like' buttons are now easier to access for both left and right-handed users. Source: own elaboration.

asked to tap the screen on different locations using their right-hand thumb only. The study took into account the limited range of movement of the thumb on the screen and the different locations it can reach on small touchscreen devices. Researchers concluded that a 9.2 mm size is large enough to reach single targets on mobile device apps.

*Guo et al. (2016)* explored the right matching between object location and user handedness, asking volunteers to tap on like and dislike buttons that were located on either side of display. For a right-handed user, the like button was placed on the right side of the screen, making it easily accessible. The dislike button was located further to the left of the screen, making it more difficult to access. Researchers recognized that there are several handedness dependant elements in the user interface of mobile devices that must be configured in a way different than in their desktop counterparts.

A similar study by *Perry & Hourcade (2008)* quantifies the performance change produced when the users deal with interfaces for mobile devices not designed to match the interaction requirements of their handedness. In the study, half of the volunteers were asked to execute

several tasks using their non-preferred hand (left or right). As expected, the study showed that users operating the device with their non-preferred hand were slower and less accurate. This effect has more impact when the target is located on the side opposite to the operating hand.

These studies also showed that the bigger the target, the easier, more accurate and quicker it was for the users to reach it. *Perry & Hourcade (2008)* reported that this phenomenon happened even whenever the non-preferred hand was used. The effect is supported by Fitts's Law (*Fitts, 1954*), which states that the time required to point to a target by a hand or a finger can be estimated as a logarithmic function of the distance and the size of the target. The bigger the target is or the closer it is to the thumb, the faster it will be to reach it. This reasoning can also be applied to the accuracy required to reach the target. Fitts's Law shows the need to decrease the distance between the thumb and the interactive targets in order to achieve a comfortable one-handed interaction experience.

If handedness can be properly determined, it would be possible to mitigate the associated usability and accessibility issues through proper user interface customization strategies.

Determination of the operating hand and posture has been studied by several researchers who proposed different algorithms. These algorithms are mostly based on the analysis of the areas where the users tap on the screen. Separate studies by *Goel, Wobbrock & Patel (2012)* and *Löchtefeld et al. (2015)* combine the detection of changes in the size of the touching area with the screen location where touching is produced to infer the operating hand. The algorithms assume that a more frequent contact is done on the screen side further away from the thumb (the easier to reach zone). Using a similar approach, *Seipp & Devlin (2015)* determined that the size of the touch zone depends on the finger used to operate the device. This area is much larger when using the thumb than when using the index finger. They concluded that a horizontal touch offset over the center of a button was a strong indicator of the operating hand. *Goel, Wobbrock & Patel (2012)* also included the study of touch trace analysis as a relevant factor to detect the operating finger. Their heuristic-based prototype assumes that the thumb-based traces consistently create an arc in contrast to the index finger where this consistency is not found. The algorithm analyses the x-displacement of the traces recorded, biasing towards a thumb-based interaction whenever the measure is greater than 5% of the screen resolution.

*Guo et al. (2016)* designed an Android-based prototype that determines the user's operating hand and the hand-changing process, combining touchscreen trace and data provided by the device's accelerometers and gyroscopes.The evaluated trace data include speed, X and Y displacements, curvature, convex orientation and the total trace length to obtain an accuracy of 95.6%. The study included only 14 volunteers who participated under the supervised conditions of a usability lab. They were asked to swipe the operating finger in each one of four possible directions: left, right, up and down, recording their actions, so user spontaneous interaction behaviour on a free context was ignored.

Another study by *Löchtefeld et al. (2015)* bases the user's operating hand detection algorithm on a PIN and password phone-unlocking system. Researchers discovered that when right handed users tried to unlock their phones with their (right) thumb, they showed a tendency to swipe from center to right. Left handed, however, tended to swipe their thumbs

to the left area of the display. Although the researchers achieved high rates of accuracy in their study, the number of users observed was only 12 and all of them where right-handed. A similar study based on data gathered from a PIN/password phone-unlocking process was designed by *Buriro et al. (2016)*. They combined data about touching zones with information provided by accelerometers and gyroscopes. They managed to determine the user's operating hand with a high level of accuracy and at the same time, they inferred information about the gender and age of the users. Unfortunately, both algorithms are based on a heavily domain dependent task (phone-unlocking) so it may be hard to extrapolate these results to other domains. These algorithms can neither be used without installing platform dependant software in the target mobile devices. They also force users to execute specific phone-unlocking tasks to update information about the user's operating hand. Those PIN based phone-unlocking tasks have been largely superseded by footprint recognition or by face recognition in modern mobile faces.

The research methodology employed in most of these studies is based on a similar approach. Researchers ask volunteers to tap or swipe over specific (restricted) areas of the screen in order to gather relevant data to be used by the algorithms. Information about hot areas is frequently complemented with accelerometer and device orientation readings coming from the gyroscopes installed on the mobile devices. This involves a strong device dependency, since specific hardware (accelerometer and gyroscope sensors) is required. In this regard a platform dependant development (Android, iOS, etc.) is required to access the information provided by the sensors, as this data is crucial to infer the operating hand. The main drawback of this approach is that mobile web-based applications cannot access this information right from the web browser, as they need the explicit user permission. In addition, not all mobile browsers offer this functionality (*Mozilla, 2019*).

Although several of the mentioned studies succeed in determining the user's handedness, obtaining moderate to high levels of accuracy, they were not able to do it implicitly. That is, they were not able to determine the user's handedness through the (stealth) observation of the users' spontaneous behavior while they browse freely through the web with their mobile devices. In the mentioned studies, small number of users were asked to execute specific actions (such as swiping their fingers in the horizontal or vertical directions or unlocking their phones using a specific finger) that were not directly related to those required to execute the users' everyday tasks. Therefore, these actions were unfamiliar to the users. Buttons and other kinds of relevant interactive web elements, like scroll controls, were not neither in the tests. *Kaikkonen et al. (2005)* showed that, under such kind of controlled interaction environment, users show a strong bias to adapt their behavior to the one expected by the observers. Thus their behavior may be different if they do similar tasks in their natural own environment.

All these solutions require users to perform unfamiliar actions that they don't usually execute in their everyday interaction environments. This makes the handedness categorization process explicit rather than implicit, so the predicting algorithms are hard to be used at all in real web scenarios if they are intended to detect the handedness of anonymous users browsing the web.

Therefore, the research hypothesis we try to address is whether it is possible to reach a level of accuracy similar to those obtained by the above mentioned studies but, using implicit detection tactics instead. That is, trying to infer handedness through the stealth observation of the spontaneous behavior of users while they freely navigate through the web. All the data required is supposed to be captured by the web browser itself, without requiring access to the mobile device sensors (eg. accelerometers or gyroscopes).

## DESIGN OF THE STUDY

In order to validate the previously described hypothesis a study was conducted, composed by the following phases:

1. Workspace design. A generic E-Commerce website prototype was developed to be freely and spontaneously explored by the volunteers participating in the study.
2. Data gathering. Software agents were deployed on the prototype to observe by stealth, the actions performed by the users.
3. Selection of Subjects. A volunteer recruiting process was done on an E-commerce user target population obtaining a probability sample of 174 volunteers.
4. Variable selection. The coordinates of the click and scrolls operations, the scroll displacement as well as the mean slope of the thumb sub-traces gathered during the navigation sessions were considered as the dependent variables.
5. Statistical Methods. As it happens in the target population, the sample was highly unbalanced (there were many more right-handed users than left-handed users) so resampling techniques were required. To increase the Information Gain Ratio, feature selection techniques were applied too in order to discard attributes that added noise.
6. Algorithm Evaluation. The processed sample was used to train and to evaluate a considerable number of classifiers. A ranking was obtained based on several accuracy markers.

### Design of the test

To simulate a real mobile web environment, a web application was developed containing a series of tests which had to be completed using two types of interaction tasks: scrolling and tapping. To avoid the use of external non standard libraries, the native TouchEvent API for JavaScript was used. This API, shared by the most popular web browsers, allows developers to detect when a user initiates or finishes a touch trace as well as to gather information during each touch trace. The web application would be later distributed through social networks to gather data from a large user pool.

Figure 5 shows the flow of the experiment and the type of data recorded in each of the pages. To initiate the test, the users first had to click on a ''start'' button. They would be then presented with instructions and the context of the test. To continue, they were asked to press another button. Then, the test began allowing users to freely navigate through a web document in order to find an object that was located at the bottom of the document. The object represented the classic Call To Action found in modern and E-commerce user interfaces. When the users clicked on that object, they were taken to the instructions page for the second test. Again, they were required to press a button to continue the tests. The

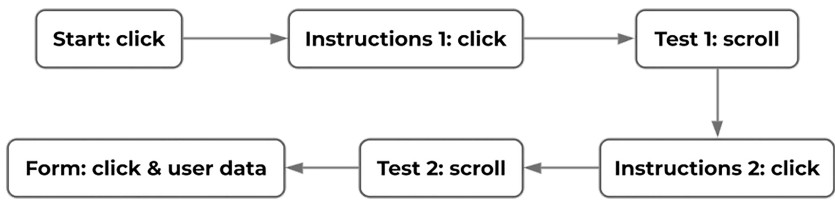

**Figure 5** **Transition diagram for the experiment and type of data recorded in each step.** Source: own elaboration.

second test involved finding an uppercase letter within a body of text in a web document. When the users clicked on the letter, they were taken to the last page where a web form asked users to indicate the operating hand and posture used during the tests. Their age, gender and some other relevant information about the users was also collected in that web form.

## Data gathering

One of the issues found in described studies was the low number of test subjects: 14 (*Guo et al., 2016*), 32 (*Azenkot & Zhai, 2012*), 12 (*Löchtefeld et al., 2015*), 14 (*Seipp & Devlin, 2015*, and (*Goel, Wobbrock & Patel, 2012*). A major goal in this study was to gather information from a large number of users to better simulate the data that would be gathered from a real web application with many diverse users.

The data gathering phase consisted of a 3-day period in which the site was made public https://lateral.herokuapp.com/en and shared through social media (Facebook, Twitter and WhatsApp). The goal of this approach was to recruit users who had some experience in browsing the Internet with mobile devices as well as obtaining from them abundant real user data for the study.

Three main categories of data were collected: data related to button clicks, data related to the user's swiping behavior and data provided by the user. The testing web site included a total of seven pages. Three pages had buttons that spanned the width of the screen: the start page, the instructions pages, and the form. The position where the users clicked on each button was recorded, measuring each click's coordinates.

Additionally, two pages were specifically designed so that mobile users were required to scroll up and down. Both pages required looking for a Call to Action object which was located at the bottom of the page and out of view. This way, users unwittingly generated scroll data while focusing on the search task. This idea is represented in Fig. 6, where "Call to action" represents the position of the object to be found. Scroll data was recorded as a collection of points, each with X and Y coordinates.

The button and scroll variables recorded were based on previous studies which showed the importance of the curve formed by a finger swipe (*Guo et al., 2016*; *Goel, Wobbrock & Patel, 2012*) and the X-position of a button click (*Seipp & Devlin, 2015*). Finally, the users were asked to fill a form to provide information about the hand used to perform the experiment, as well as some other information to serve as sample description: gender, age, weekly computer usage (hours/week), and device type.

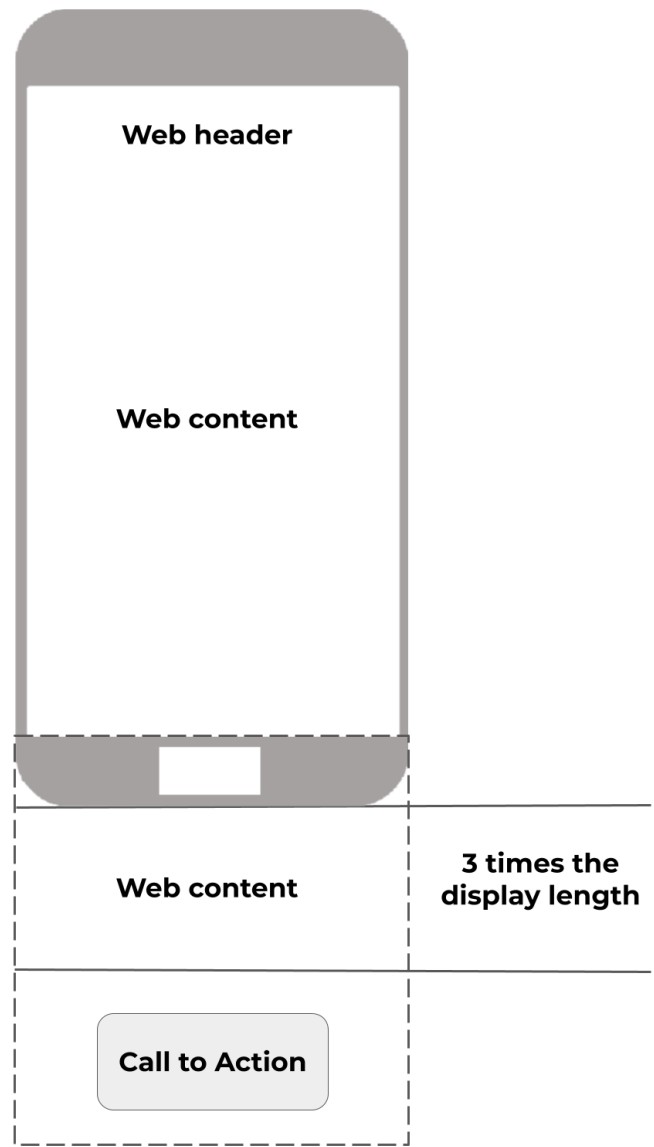

**Figure 6** Two scroll tests were designed specifically to allow users to scroll through the web document. Source: own elaboration.

The main goal behind this experimental design was to prevent subjects from being aware of the object of the study. To achieve this, the tests were designed as games, and the users were simply asked to use the same hand and posture throughout the experiment while looking for the items on the screen. Thus, they did not know that their scrolls and clicks were being recorded. Furthermore, the experiment was accessible through a website and open to the public. The raw data captured was directly stored for future analysis.

## Analyzed sample

The experiment yielded data from 174 voluntary users. Out of these, 35 completed the experiment with their left hand, whereas 139 performed it with the right hand, as can

**Table 1  Sample distribution.**

| Variable | Category | Occurrences |
|---|---|---|
| Hand | Left | 35 |
| | Right | 139 |
| Age | [15-25] | 81 |
| | (25-35] | 9 |
| | (35-45] | 34 |
| | (45-55] | 41 |
| | (55-65] | 7 |
| | (65-75] | 2 |
| Gender | Male | 98 |
| | Female | 76 |

be seen in Table 1. This meant that the resulting sample was highly unbalanced, which might have caused the machine learning algorithms to bias towards the majority class, providing low classification power for the minority class. Thus, as will be explained later, it was required to apply a re-sampling technique to balance the classes before training the classifiers (*García et al., 2007*).

## VARIABLES OF THE STUDY

The gathered data for the scrolls was filtered to separate each touch trace into a set of sub-traces. This step was necessary because a single trace could be the result of users swiping up and down multiple times without lifting their finger. Therefore, every time the trace changed direction (upward to downward or downward to upward), a new sub-trace was created to preserve the validity of features such as start and end points, slope, and maximum and minimum X-positions. Next, each sub-trace was passed through a secondary filter to ensure it contained at least two touch-points to provide meaningful results, since a sub-trace with only one touch-point would represent a user's misclick when scrolling. The gathered data for the clicks did not require any such filtering. Finally, the resulting set of features is comprised of:

### Click X and Y positions

The mean and median were calculated for the set of X coordinates and the set of Y coordinates from each of the user's clicks. Each X and Y coordinate is relative to the button being clicked. In the study by *Guo et al. (2016)*, the only data considered was based on scrolls. In this study, click data was included to provide a more accurate description of the user's interactions with a mobile application.

### Scroll X and Y positions

For each user, the positions of each point recorded, along their scroll sub-traces, were examined. From this data, the means were calculated for the maximum, minimum, initial and overall X-values for each of those sub-traces. Lastly, the standard deviation and median were calculated for the user's set of X and Y-values from all of their sub-traces.

## Scroll X and Y displacements
The displacements from the maximum to the minimum X and Y values along each scroll sub-trace were calculated and then averaged over the set of sub-traces for each user.

## Mean slope
The mean slope for the user's sub-traces was calculated by computing the mean of the slopes from the starting point to the end point of each sub-trace (Eq. (1)). *Guo et al. (2016)* included more curve shape descriptors in their algorithms (Root Mean Squared Error, Maximum Curvature, Average Curvature and Curve Convex Orientation). Although these measures might provide a more exhaustive description of the curve, they increase the computational workload. In contrast, as we will evaluate later, the slope of a curve is a simpler operation which still serves its purpose as a trace descriptor.

$$m = \frac{\Delta y}{\Delta x} = \frac{y_n - y_0}{x_n - x_0} \tag{1}$$

## STATISTICAL METHODS
Several pre-processing techniques were applied to the filtered data to balance the classes and remove irrelevant and redundant information from the feature set.

### *Resampling*
For the purpose of this study, no preference was given in the prediction model for either the left or the right hand. However, as seen in the sample distribution, more subjects performed the test using their right hand, resulting in few left-hand operation examples, which produced an unbalanced data-set. Furthermore, the natural distribution in a classification problem seldom produces the best-performing classifier (*Weiss & Provost, 2001*). Therefore, a re-sampling technique was applied, randomly oversampling users from the ''left'' class with replacement until both classes were balanced. The resulting data-set contained the original 139 right-hand examples and the re-sampled 139 left-hand examples. This method was used due to its good performance in solving the class imbalance problem (*Japkowicz & Stephen, 2002*; *Batista, Prati & Monard, 2004*).

### *Feature selection*
Further data pre-processing was applied to discard any attributes that might add noise to the classification. The complete set of features was evaluated and ranked using the Information Gain Ratio (IGR) as an attribute selection method.

The results for the Information Gain Ratio evaluation can be found in Table 2. The average X-position of the clicks was found to be the most informative feature. Average scroll X-position, along with the average maximum and minimum points of the scrolls, also showed a high degree of information gain towards the classification. This supports this study's approach for a mixed classification system, combining click and scroll data to better predict the user's operating hand. However, four features provided no useful information for this classification problem, each having a ranking of 0 after the evaluation. For the

**Table 2 Information gain ratio attribute selection ranking.**

| Feature | IGR | Feature | IGR |
|---|---|---|---|
| Mean X (clicks) | 0.487 | Mean X (scrolls) | 0.432 |
| Mean Minimum X (scrolls) | 0.395 | Median X (clicks) | 0.384 |
| Mean Start X (scrolls) | 0.323 | Median X (scrolls) | 0.305 |
| Mean Slope | 0.178 | Mean Y Displ. (scrolls) | 0.148 |
| Mean Y (clicks) | 0.147 | Median Y (clicks) | 0.117 |
| Median Y (scrolls) | 0 | Mean X Displ. (scrolls) | 0 |
| Std. Dev. X (scrolls) | 0 | Std. Dev. Y (scrolls) | 0 |

standard deviation measures, this might mean that both left and right-handed users are just as consistent with the areas of the screen they use.

## ALGORITHM EVALUATION

It is possible that, as each individual user generates more and more touch data during their navigation session, the scrolling manner could be affected in some way during that session so there could exist specific patterns present in their first scroll actions that may vary in later scrolls actions. Furthermore, it would be interesting to gauge whether the predictive power of the algorithms under evaluation varies when considering a specific number of scrolls actions and whether a different classifier could perform better than the proposed one based on that figure.

For this reason, the classification models were finely tuned for the prediction of the operating hand considering a specific $n$ number of scroll actions as the main parameter. The initial goal was to obtain faster, but slightly less accurate classifiers, created from training data coming from a very few scroll actions done by users who just arrived at the web document. But the goal also pursued to define slower but more accurate classifiers created with more touch trace data, coming from users who spent more time exploring the website.

In order to perform individual scroll analysis, the original sample was split into several sub-samples. Each of these sub-samples contains the features previously described for all the recorded scrolls actions from only 1 scroll action up to $n$. The re-sampling method previously described was consequently applied to each sub-sample. That is, the minority class examples were over-sampled until both classes were balanced.

The amount of examples in each sub-sample decreases as the number of recorded scroll actions grows larger. This occurs because some users were faster when finding the Call to Action objects in the test, thus completing it without generating as much scroll data as other users.

A considerable number of classifiers were trained and evaluated. Some of them were chosen based on their success in previous studies, such as Random Forest, used by *Guo et al. (2016)* and *Seipp & Devlin (2015)*, Multi-Layer Perceptron, used by *Guo et al. (2016)* or C4.5, used by *Guo et al. (2016)*, *Goel, Wobbrock & Patel (2012)* and *Seipp & Devlin (2015)*. Sequential Minimal Optimization (SMO) (*Platt, 1998*) is one of the most

**Table 3** Classification results for $n = 1$.

| Algorithm | TPR | Incorrect | F-Measure | AUROC |
|---|---|---|---|---|
| AdaBoost Decision Stump | 98.16 | 2.00 | 0.98 | 0.99 |
| AdaBoost PART | 98.01 | 2.16 | 0.98 | 0.99 |
| Random Forest | 97.80 | 2.40 | 0.99 | 1.00 |
| PART | 96.98 | 3.28 | 0.97 | 0.97 |
| C4.5 | 96.85 | 3.43 | 0.97 | 0.97 |
| k-Star | 96.75 | 3.54 | 0.97 | 0.99 |
| KNN | 96.68 | 3.62 | 0.97 | 0.96 |
| MLP | 95.32 | 5.10 | 0.95 | 0.95 |
| Logistic Regression | 95.11 | 5.32 | 0.95 | 0.97 |
| SMO | 94.34 | 6.16 | 0.94 | 0.94 |
| Naive Bayes | 84.32 | 17.05 | 0.83 | 0.64 |

popular algorithms for training Support Vector Machines (SVMs). It was not used in the aforementioned studies, but was chosen for its wide applicability in pattern recognition and classification problems (*Naveed et al., 2019*). Others used by other researchers in similar studies, such as Naive Bayes (*Guo et al., 2016*) and K-Nearest Neighbors (*Guo et al., 2016*; *Löchtefeld et al., 2015*; *Seipp & Devlin, 2015*) were also included. Furthermore, other algorithms were tested and the best-performing ones were included into the experimental set of classifiers. These include K-Star, PART (partial decision tree), Adaptive Boosting with Decision Stumps, Adaptive Boosting with PART, and Logistic Regression.

The classifiers were evaluated by using a random 66% split on the data and averaging the results over 200 iterations. Table 3, shows the results obtained for $n = 1$, that is when only the first scroll action done by the users in the web site is considered. As can be seen by considering only the first scroll action done by a user, the best model (AdaBoost Decision Stump) achieves on average a 98.16% TPR, classifying on average 2 examples incorrectly.

In a practical application, this approach would provide a quick prediction as the user starts scrolling, in exchange for some loss in the quality of the said prediction. As the user generates more touch trace data, the classifiers learn from the new information while still considering the previous one. This means that the classification power increases and predictions are more accurate and robust. This is indeed confirmed by the results shown in Table 4 for $n = 2$ where the best classifier (AdaBoost PART) achieves a 98.94% TPR and the prediction is incorrect only for one instance. The corresponding tables for values of n from 3 to 6 are included in Tables A1, A2, A3 and A4 in the Appendix (see appendix).

Finally, the best classification was achieved for $n = 7$ by partial decision trees (PART), classifying incorrectly, on average, only 0.33 of the 42 testing instances in this sample on average (see Table 5). This means that in most iterations the classifier provided perfect results, predicting the user's operating hand with 100% accuracy (see Table 5).

The results obtained in these tests demonstrate that when considering scrolls actions individually and adding new information as the user generates touch trace data, the classifiers can provide highly educated predictions from the moment the user starts scrolling and even more accurate ones with as little as 7 interface interactions.

**Table 4** Classification results for $n = 2$.

| Algorithm | TPR | Incorrect | F-Measure | AUROC |
|---|---|---|---|---|
| AdaBoost PART | 98.94 | 1.00 | 0.99 | 1.00 |
| AdaBoost Decision Stump | 98.88 | 1.06 | 0.99 | 1.00 |
| Random Forest | 98.76 | 1.18 | 0.99 | 1.00 |
| KNN | 98.21 | 1.69 | 0.98 | 0.98 |
| k-Star | 97.95 | 1.94 | 0.98 | 0.99 |
| C4.5 | 97.94 | 1.95 | 0.98 | 0.98 |
| PART | 97.60 | 2.26 | 0.98 | 0.98 |
| Logistic Regression | 94.30 | 5.39 | 0.94 | 0.98 |
| MLP | 94.00 | 5.67 | 0.94 | 0.97 |
| SMO | 93.78 | 5.87 | 0.93 | 0.94 |
| Naive Bayes | 92.32 | 7.26 | 0.92 | 0.95 |

**Table 5** Classification results for $n = 7$.

| Algorithm | TPR | Incorrect | F-Measure | AUROC |
|---|---|---|---|---|
| PART | 99.92 | 0.33 | 0.99 | 0.99 |
| AdaBoost Decision Stump | 99.28 | 0.30 | 0.99 | 0.99 |
| C4.5 | 99.19 | 0.33 | 0.99 | 0.99 |
| Logistic Regression | 98.70 | 0.54 | 0.99 | 1.00 |
| AdaBoost PART | 97.75 | 0.93 | 0.98 | 0.99 |
| Random Forest | 97.62 | 0.99 | 0.98 | 1.00 |
| SMO | 95.46 | 1.88 | 0.95 | 0.95 |
| k-Star | 95.26 | 1.96 | 0.96 | 0.98 |
| MLP | 95.06 | 2.05 | 0.95 | 0.97 |
| KNN | 94.98 | 2.08 | 0.95 | 0.94 |
| Naive Bayes | 93.60 | 2.65 | 0.92 | 0.98 |

For comparison to the 99.92% obtained, it is remarkable that *Guo et al. (2016)* achieved a precision of 95.6% on data gathered from with an Android-specific implementation and fewer test subjects and *Löchtefeld et al. (2015)* attained a TPR of 98.5% by gathering data during the phone unlocking process, including gyroscope and accelerometer readings. Although their systems achieved similar results compared to ours, they are not always applicable, such as in mobile web applications where the phone unlocking process is rarely used. Furthermore, these approaches require access to the device's sensors, whereas the approach proposed in this study is completely sensor-independent and applicable for any device with a touchscreen.

The improved results obtained in this study are probably due to the combination of click and scroll data, which provide more information for the classifiers than using only one of them. Furthermore, the addition of several descriptors for the values of the scrolls, such as the median or the starting points, and the inclusion of the slope descriptor might also have had a positive influence on the classification power.

## LIMITATIONS

The age distribution of the sample comprised users from 15 to 74 years. While the most recurrent user age groups are those from 15 to 25 and from 35 to 55 years, only 2 of the subjects were in the range from 65 years onward, and no subject was younger than 15. This sample distribution is consistent with the target population of e-commerce users and consequent to the volunteer recruit strategy employed, as users belonging to those age groups tend to use these interaction scenarios more frequently than older or younger users (*Hutchinson, 2017*).

Although this sample distribution reinforces the internal validity of the study, it weakens its external validity in specific age ranges as the study cannot draw conclusions concerning user profiling of children and/or elderly users.

Regarding gender distribution, 44% of the test subjects were female, in comparison with the approximately 50% found in the global population (*World Bank, 2017*). This slight difference may be due to the method of distribution of the experiment. Nevertheless, this study's gender ratio remains representative of the global population.

Additionally, the data in this research was gathered from Spanish and English-speaking users. These languages belong to the Western culture and share the same writing direction (from left to right), among other common characteristics. As a result, the profiling model may not be extensible to other cultures, specifically to the users of the majority of the Semitic languages (like the Arabic, the Amharic, etc.) which are written from right to left.

## CONCLUSIONS AND FUTURE WORK

The goal of this research was to implicitly determine the operating hand of a mobile device user in an E-commerce web application. Previous studies had mostly focused on sensor-based solutions, with few test subjects. The study by *Guo et al. (2016)* was the closest we found to our goal (F-Measure value of 0.956), although in their approach the user is required to do specific tasks on an Android-based platform, whereas our user can freely use their preferred device to navigate through a web document.

Our findings suggest that the best classification device is a partial decision tree trained using a combination of features gathered through the evaluation of button clicks and scroll traces from 174 voluntary users, detecting the user operating hand with an TPR value of 99.92. However, we must point out that whenever it is required to get a quick classification based on analysis of very few user interactions ($n = 1$, $n = 2$...), the algorithms based on boosting techniques (such as AdaBoost Decision Stump or AdaBoost PART) are the ones that perform better.

To the best of our knowledge, this approach is the first to explore and propose a solution for operating hand detection in mobile web applications using only data gathered from the touchscreen when the user spontaneous carries out web browsing tasks. Although the tasks studied are focused on click and scroll up/down, the relatively high level of classification accuracy obtained (99.6%), ruled us out to explore the effect of other, no so common web browsing tasks, such as for example scroll left/right or zoom in/out.

Unlike other proposals covered in this document, the proposed solution is based on implicit determination of the user's handedness based on stealth observation of natural interaction tasks. It does not require users to perform uncommon tasks in order to determine their handedness. It neither requires to install platform dependant software on the target mobile device, neither the use of sensors, avoiding calibration an extra battery consumption.

E-commerce web application developers can make use of these findings to detect the handedness of anonymous users visiting their web sites after observing their natural (spontaneous) interactions for a few time in stealth mode. The proposed algorithm is able to detect the user's handedness with moderate-high level of accuracy, thus enriching the user model required by their applications through the user interface personalization process.

Although discussion about how this personalization process may be implemented is beyond the scope of this paper, E-commerce applications may improve their user experience providing custom interfaces for left and right-handed users. Hence, the personalization process may provide accessibility-based solutions to the specific user interaction requirements of each kind of user.

The data gathering agents developed in this study can be deployed in the target web applications to feed inference algorithms running on the server-side. Boosting-based algorithms, like AdaBoost Decision Stump or AdaBoost PART, can be used to perform a first and quick classification round based on one or two user interactions followed by the execution of partial decision trees (once more data from ore user interaction is gathered) to get a more accurate classification. These algorithms can classify the visiting anonymous users, updating these results in the user model. This information may be used in several ways to adapt the user interface (eg. through the use of customized CSS), hence increasing the overall user experience.

As mentioned in the limitation section, the way the users creates sub-traces while they freely perform scrolls operations during their navigation sessions may be influenced by other factors rather than their handedness, such as the language spoken. This study revealed a few interesting future research topics. One of these involves studying the influence of user's culture. Activities such as swiping and clicking might be biased by cultural aspects like the writing direction of the user's language. Thus, studying cultural differences performing these interactions might improve this research, widening the applicability of the solution.

As mentioned in the limitation section, activities such as swiping and clicking might be biased by cultural aspects like the writing direction of the user's language. The data samples obtained for this study were based on languages written from the left to the right only. Thus, studying cultural differences behind these interactions might improve this research, widening the applicability of the solution to, for example, languages written from the right to the left.

## ACKNOWLEDGEMENTS

We thank Brian Zylich (Dept. of Computer Science, Worcester Polytechnic Institute, USA) and José Florez (Dept. of Electrical & Computer Engineering, Universidad del Turabo,

Puerto Rico) for their collaboration in the data gathering process at the early stages of this research.

# APPENDIX

Classification results of n from 3 to 6.

**Table A1  Classification results for $n = 3$.**

| Algorithm | TPR | Incorrect | F-Measure | AUROC |
|---|---|---|---|---|
| AdaBoost PART | 98.54 | 1.20 | 0.99 | 0.99 |
| AdaBoost Decision Stump | 98.30 | 140 | 0.98 | 0.99 |
| Random Forest | 98.10 | 1.56 | 0.98 | 1.00 |
| C4.5 | 97.92 | 1.72 | 0.98 | 0.98 |
| PART | 97.81 | 1.80 | 0.98 | 0.98 |
| k-Star | 96.96 | 2.50 | 0.97 | 0.99 |
| KNN | 96.41 | 2.91 | 0.96 | 0.97 |
| Logistic Regression | 95.92 | 3.36 | 0.96 | 0.98 |
| MLP | 95.03 | 4.09 | 0.95 | 0.97 |
| SMO | 94.32 | 4.67 | 0.94 | 0.94 |
| Naive Bayes | 91.96 | 6.61 | 0.92 | 0.95 |

**Table A2  Classification results for $n = 4$.**

| Algorithm | TPR | Incorrect | F-Measure | AUROC |
|---|---|---|---|---|
| Random Forest | 98.74 | 0.89 | 0.99 | 1.00 |
| AdaBoost PART | 98.67 | 0.94 | 0.99 | 0.99 |
| AdaBoost Decision Stump | 98.10 | 1.34 | 0.98 | 0.99 |
| k-Star | 97.84 | 1.53 | 0.98 | 0.99 |
| C4.5 | 97.59 | 1.70 | 0.98 | 0.98 |
| PART | 97.51 | 1.76 | 0.97 | 0.98 |
| Logistic Regression | 97.11 | 2.05 | 0.97 | 0.99 |
| SMO | 97.00 | 2.11 | 0.97 | 0.97 |
| MLP | 96.23 | 2.66 | 0.96 | 0.98 |
| Naive Bayes | 94.69 | 3.76 | 0.95 | 0.97 |

**Table A3  Classification results for $n = 5$.**

| Algorithm | TPR | Incorrect | F-Measure | AUROC |
|---|---|---|---|---|
| k-Star | 98.60 | 0.83 | 0.99 | 0.99 |
| KNN | 98.41 | 0.94 | 0.98 | 0.98 |
| AdaBoost Decision Stump | 97.88 | 1.26 | 0.98 | 0.99 |
| C4.5 | 97.54 | 1.46 | 0.98 | 0.98 |
| AdaBoost PART | 97.68 | 0.98 | 0.98 | 0.99 |
| PART | 97.46 | 1.51 | 0.97 | 0.98 |
| Logistic Regression | 97.22 | 1.64 | 0.97 | 0.98 |
| Random Forest | 97.14 | 1.69 | 0.99 | 0.99 |
| MLP | 96.22 | 2.23 | 0.96 | 0.97 |
| SMO | 95.82 | 2.48 | 0.96 | 0.96 |
| Naive Bayes | 92.87 | 4.22 | 0.93 | 0.97 |

**Table A4  Classification results for $n = 6$.**

| Algorithm | TPR | Incorrect | F-Measure | AUROC |
|---|---|---|---|---|
| C4.5 | 99.16 | 0.42 | 0.99 | 0.99 |
| PART | 99.16 | 0.42 | 0.99 | 0.99 |
| AdaBoost Decision Stump | 98.75 | 0.63 | 0.99 | 0.99 |
| AdaBoost PART | 97.84 | 1.08 | 0.98 | 0.99 |
| Logistic Regression | 97.57 | 1.22 | 0.97 | 0.99 |
| Random Forest | 97.19 | 1.42 | 0.97 | 1.00 |
| MLP | 96.33 | 1.84 | 0.96 | 0.98 |
| k-Star | 96.07 | 1.98 | 0.96 | 0.98 |
| KNN | 95.36 | 2.34 | 0.96 | 0.96 |
| SMO | 94.79 | 2.62 | 0.95 | 0.95 |
| Naive Bayes | 92.05 | 4.00 | 0.92 | 0.97 |

### Funding

This work was funded by the Department of Science, Innovation, and Universities (Spain) under the National Program for Research, Development, and Innovation (project RTI2018-099235-B-I00) and the National Science Foundation under grants No. 1458928 and No. 1645025, an REU Site on Ubiquitous Sensing. There was no additional external funding received for this study. The funders had no role in study design, data collection and analysis, decision to publish, or preparation of the manuscript.

### Grant Disclosures

The following grant information was disclosed by the authors:
The Department of Science, Innovation, and Universities (Spain) under the National Program for Research, Development, and Innovation (project RTI2018-099235-B-I00).
The National Science Foundation: No. 1458928, No. 1645025.

REU Site on Ubiquitous Sensing.

## Competing Interests

The authors declare there are no competing interests.

## Author Contributions

- Carla Fernández conceived and designed the experiments, performed the experiments, analyzed the data, performed the computation work, prepared figures and/or tables, authored or reviewed drafts of the paper, and approved the final draft.
- Martin Gonzalez-Rodriguez conceived and designed the experiments, performed the experiments, analyzed the data, prepared figures and/or tables, authored or reviewed drafts of the paper, and approved the final draft.
- Daniel Fernandez-Lanvin conceived and designed the experiments, performed the experiments, analyzed the data, authored or reviewed drafts of the paper, and approved the final draft.
- Javier De Andrés conceived and designed the experiments, analyzed the data, performed the computation work, authored or reviewed drafts of the paper, and approved the final draft.
- Miguel Labrador conceived and designed the experiments, authored or reviewed drafts of the paper, and approved the final draft.

## Data Availability

The raw data and the Python script used to aggregate the data and to perform basic filtering are available in the Supplemental Files.

## Supplemental Information

Supplemental information for this article can be found online at http://dx.doi.org/10.7717/peerj-cs.487#supplemental-information.

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
