# Peer review of "Implicit detection of user handedness in touchscreen devices through interaction analysis"

_PeerJ Computer Science, doi:10.7717/peerj-cs.487_

## Round 0.1 · original submission · Minor Revisions

Dear Authors,
Please carefully revise the article and you are welcome to resubmit. Thank you

Reviewer 1 ·

Basic reporting

This paper presents an interesting and relevant study about the operative used user’s hand when using a mobile application. It is well written and easy to read. This could help on issues related to the design of user interfaces (assuring usability and accessibility) of mobile application, and also in their implementation in order to avoid the use of mobile sensors that impact un battery consumption and other related features. To that end, a model based on the use of machine learning techniques to classify users implicitly and dynamically according to their operational hand was applied, obtained a high level of accuracy which makes the result really relevant for the community of software engineering, human-Computer Interaction and mobile design. The paper presents an up-to-date and good background on the topics used on the study and related studies performed in the same domain. The study was performed with a large set of participants (174), following good experimental procedures and statistical analysis. All terms and concepts used in the paper are supported by references.

Experimental design

This paper is completely whiting the scope of PJeer Computer Science.
Section 3 presents the design study from its definition (section 3.1) to the execution. It clears presents from section 3.2 how the procedures from section 3.1 was applied, all variables analyses, collected data, how this data was statistical analyzed and the conclusions obtained suppirted by the data. However, section 3.1 (Design of the test) limits to describe the test procedure (step by step) performed by the participants. It is really important for the replicability of the study, to present in this section the complete research protocol of the study establishing the plan followed in the experiment, that is : the context selection, the hypothesis formulation, variables selection (independent and dependent variables, selection of subjects (if it’s a a probability or a non-probability sample), instrumentation, the statistical test planned to analyze the results with justification, and validation. All this information is presented as the results are described in the following sections, but in a dispersed way. That means the experiment really followed all rules of empirical studies which gives confidence for the reader and the researches that intend to use the results from this paper. My suggestion is just to value the study and allow others to not only replicate it but use the protocol as a basis for other work. Moreover, having a complete research protocol previously with a complete view of the study helps in the understanding of the execution of the study and analysis of the results.

Validity of the findings

The paper presents very relevant results for the community that work on the design of user interfaces for mobile applications. The conclusion is well stated. The goal of the study is recalled and the performed procedures to address the goal is summarized. The main results and originality of the study are also highlighted.
The conclusion (section 7 ) is followed by another section (section 8) that presents the limitation of the work and future works. In fact, the future works is limited to the last paragraph. I suggest that future works be transferred for the section of conclusion and more explored. Moreover, that the limitations be presented before the conclusion. In other words, I suggest that section 8 be divided and limitations be presented before the conclusion section to better discuss the validity of the sudy.

The validity of the study is discussed in the limitation section explicitly explaining some results and the conditions to be replied, However, as an experimental experiment, threats of validity should be analyzed to better discuss the limitations. The way it is presented does not really address all validity criteria (internal validity, construct validity, external validity and conclusion validity) of an empirical study. A good suggestion about the threats to validity in experimentation can be foun in the book of C. Wohlin, P. Runeson, M. Höst, M. C. Ohlsson, B. Regnell, and A. Wesslén, Experimentation in software engineering. Springer Science & Business Media, 2012.

Additional comments

It is a very interesting study with impact results for research community and industry that development of mobile application)

Reviewer 2 ·

Basic reporting

This is an interesting article about the analysis of users interaction on touchscreen. The motivation of the paper is significant as well as the structure is well written. The experiment design of the paper is good as well as the verification also shown very good result.

Experimental design

I suggest the author can describe more about the connection between different phase of the experiments.

Validity of the findings

no comment

---

## Round 0.2 · accepted · Accept

Dear authors

Thanks for revising the paper according to reviewer comments.

Reviewer 1 ·

Basic reporting

This reviewed version answer all the issues I mentioned in the first review.

Experimental design

The authors described the design steps performed for the experiment as required in my previous review.

Validity of the findings

The authors reorganised the section to explicitly discuss the limitations of the work, the conclusions and future work.
The paper is much better.

Additional comments

Thank you very much, for the improvement in the paper.
I believe this study will be very useful for the research community.